# High-intensity eccentric hip abductor strength training improves dynamic knee valgus in a task-dependent manner in asymptomatic young women: A pilot study

Ádám Fésüs[1]*, Balázs Sebesi[1], Zsolt Murlasits[1], Patrik Ivusza[1], Judit Prókai[1], Kitty Vadász[1], Balázs Gáspár[1], Vanessza Malmos[2], Tibor Hortobágyi[1,3,4], Márk Váczi[1]

1 Faculty of Sciences, Institute of Sport Science and Physical Education, University of Pécs, Pécs, Hungary, 2 Faculty of Humanities, Institute of English Studies, University of Pécs, Pécs, Hungary, 3 Department of Kinesiology, Hungarian University of Sports Science, Budapest, Hungary, 4 Center for Human Movement Sciences, University Medical Center Groningen, University of Groningen, Groningen, The Netherlands

* fesusadi@gamma.ttk.pte.hu

## Abstract

### Introduction

Dynamic knee valgus is linked to reduced hip abduction strength, a critical factor in knee stability during unilateral movements. While interventions to reduce dynamic knee valgus often use traditional hip abduction training, many neglect the eccentric function of hip abductors, essential for controlling femoral medial translation. This pilot study compared the effects of a four-week-long eccentric vs. concentric hip abduction training on hip abduction torque, countermovement jump performance, dynamic knee valgus measured during one-legged jumping and drop landing, and determined if reductions in dynamic knee valgus correlated with increases in hip abduction torque and countermovement jump performance.

### Materials and methods

Asymptomatic, physically active female college students (n = 20, 21.3 ± 2.51 years) with dynamic knee valgus were randomized to either eccentric or concentric hip abduction strength training. Testing included maximum hip abduction torque on an isokinetic dynamometer, single-leg countermovement jumps, and single-leg drop landings analyzed with 3D motion tracking. Participants trained three times per week for four weeks, performing four sets of 10 maximal effort repetitions.

### Results

The two groups did not differ at baseline in any outcomes (all p > 0.05). Eccentric hip abduction torque improved over time (F = 39.7, p < 0.001) without a group-by-time

**Data availability statement:** The minimal
dataset required is available on the Zenodo
platform under the following DOI:https://doi.
org/10.5281/zenodo.1808373. This access
location is indicated in the Data Availability
Statement section of the manuscript, between
lines 395–398.

**Funding:** ÁF, PI, and BG were supported by
the Ildikó Kriszbacher scholarship, University
of Pécs. BS was supported from EKÖP-24-4-
I-PTE-379, funded by the Ministry of Culture
and Innovation, National Fund for Research,
Development and Innovation, under the
University Research Grant Programme EKÖP-
24-1. For the remaining authors none were
declared. The funders had no role in study
design, data collection and analysis, decision to
publish, or preparation of the manuscript.

**Competing interests:** The authors have
declared that no competing interests exist.

interaction. Dynamic knee valgus decreased during single-leg countermovement jumps (time main effect: $F = 33.5$, $p < 0.05$) and single-leg drop landings (time main effect: $F = 14.8$, $p < 0.05$). The reductions in dynamic knee valgus, measured during single-leg countermovement jumps, were greater ($p < 0.05$) after eccentric vs. concentric training (group by time interaction: $F = 5.57$, $p < 0.05$). Countermovement jump improved similarly in the two groups (time main effect: $F = 5.1$, $p < 0.05$), without group by time interaction. Improvements in hip abduction maximal torque and countermovement jump performance, and changes in dynamic knee valgus outcomes did not correlate ($p > 0.05$).

## Conclusions

High-intensity eccentric versus concentric hip abductor strength training was superior in dynamic knee valgus improvement measured during single-leg countermovement jump but not during drop landings in asymptomatic young women, while both training modalities improved single leg countermovement jump performance.

## Introduction

Dynamic knee valgus (DKV) is an abnormal lower extremity movement pattern visually characterized by excessive medial movement of the knee during weight bearing activity [1]. Dynamic knee valgus (DKV) is a significant risk factor for non-contact injuries of the anterior cruciate ligament (ACL). DKV appears as the medial collapse of the knee during tasks involving hip and knee flexion. Impaired neuromuscular control of the hip joint is most often implicated in the evolution of DKV [2]. Specifically, suboptimal activation of the hip abductor muscles and low strength levels primarily in the gluteus medius are proposed to allow excessive medial translation of the knee during weight bearing [3]. When such movements are executed repeatedly, the ACL becomes chronically exposed to high mechanical loading [4]. Both bilateral and unilateral variations of tests such as squats, countermovement jumps (CMJ), and landings are used for evaluating knee injury risks by quantifying DKV. Still, single-leg CMJs and landings seem to be the most sensitive tests to detect DKV [5]. One reason is that the squat test does not resemble the velocity profile of real-life sport movements. Another reason is that knee injury risk increases during unilateral weight bearing tasks, such as single leg jumps, cutting movements, and changes in direction [6,7]. Indeed, DKV was higher in tasks performed with one compared with two legs (landing, drop jump, squat) [8,9].

Biomechanical studies reported that DKV tends to be higher in women than men [9]. Because DKV is known to increase ACL strain, DKV is thought to be partly related to the 2–9 times higher incidence of non-contact ACL injuries in females vs. males [10–12]. It is well established that high DKV during unilateral tasks is associated with low levels of hip abduction strength in women [13–15]. A muscle-driven simulation study examined how robust human gait is to muscle weakness. The study found that 60–80% of hip abductor weakness made walking impossible, while

walking was still possible when other leg muscles showed similar weakness [16]. Such data provides strong evidence that hip abduction strength is a determinant of knee stability during movements involving unilateral stance.

While DKV is a modifiable risk factor for ACL injuries and the mechanism of how DKV evolves has been established, researchers still look for solutions to reduce DKV magnitude. One typical interventional approach to improve DKV is hip abduction resistance training. This intervention revealed either reduced [17] or unchanged [18,19] DKV. Application of biofeedback [17,20], i.e., placing motion tracking sensors on the leg, providing immediate visual feedback of DKV during exercise, or its combination with resistance training [18,21–23], are also popular approaches, even though its effectiveness remains uncertain. While certain interventions were successful in reducing DKV, in several of these studies only unilateral squat test was used for quantifying DKV [17,18,20]. Another limitation of these studies is that hip abduction strength training was conducted at a low intensity [17,18]. Finally, providing users with biofeedback can be technically challenging and costly, hence its availability is limited to biomechanics laboratories.

An important mechanism in the development of DKV is that the medial translation of the femur is controlled by eccentric activation of the hip abductors. Previous studies seemed to have overlooked this function of the hip abductors in prescribing hip abduction strength training for DKV and used traditional eccentric-concentric contractions. In such training exercises muscles are under-loaded in the eccentric phase [17,18,24]. Because DKV is controlled by eccentric contractions of the hip abductors, it is conceivable that eccentric training of the hip abductor muscles would improve DKV. This hypothesis is supported by the effectiveness of overloaded eccentric contractions on eccentric strength versus either concentric contractions or under-loaded eccentric contraction exercise training [24–26].

In addition to the increased ACL injury risk due to DKV, excessive medial position of knee joint could be biomechanically disadvantageous with respect to the direction of force application during CMJ. This is confirmed by studies showing that gluteus medius fatigue alters ankle joint kinematics [27], impairs postural control [28], and increases medial-lateral center of pressure displacement [29], contributing to an increase in risk of a loss of balance and reduced vertical ground-reaction force during single leg CMJ. In addition, women tend to land with less knee flexion during two-legged vs. one-legged landings, which might be a strategy to avoid excessive DKV [9]. Based on these studies, it seems reasonable to expect that hip abductor strength training could indirectly increase CMJ performance through optimized knee flexion during CMJ in females.

The purpose of the present study was to investigate the effects of a four-week-long eccentric vs. concentric hip abduction training on hip abduction torque, CMJ mechanical impulse, DKV measured during one-legged jumping and drop landing, and to determine if reductions in DKV correlated with increases in hip abduction torque. We hypothesized that: 1. hip abduction torque adaptation would be mode-specific to eccentric or concentric training contractions; 2. adaptation in single leg CMJ mechanical impulse would be greater after eccentric vs. concentric hip abduction training; and 3. eccentric vs. concentric hip abduction training would preferentially reduce DKV in single leg CMJ and single leg landing.

## Materials and methods

### Participants

The recruitment of participants for this study took place between 10/01/2022 and 10/03/2022. We recruited asymptomatic female college students who were physically active and free from injuries but had DKV (see below). The inclusion criteria were female gender, age between 18 and 30 years, and the presence of DKV subjectively evaluated by a physiotherapist, using the single leg squat test. Exclusion criteria were current injuries and past surgeries in the spine, hip, knee, and ankle joints, or pain of orthopedic origin. When the actual quantitative DKV measurements were preformed, all participants demonstrated greater than 8° of DKV in at least one of the two DKV tests. Considering recent literature suggesting 7–10° as thresholds for ACL injuries, we made no further exclusions [30]. Finally, 20 participants (age: 21.3±2.51 years; height: 167.8±5.76 cm; mass: 60.6±6.17 kg; exercise training history: 15.4±3.32 years) were involved in the study. Participants pursued ground-contact sports 6.3±2.42 hours per week at the club level, but none of them competed at a

national or international level. Participants received verbal and written explanations of the experimental procedures and were informed about the potential risks. They provided written consent under the principles outlined in the Declaration of Helsinki. The University Ethics Committee approved the protocol (approval number, 7961-PTE2019). No participants withdrew from the study.

## Experimental procedure

The duration of the study was five weeks, including all measurements and training in the following order: pre-testing, 4-week-long training intervention, and post-testing (Fig 1). Such a short-term intervention is known to produce changes in lower limb kinematics during sidestep cutting, a task closely associated with ACL injury risk in female athletes [31]. Participants underwent a standardized warm-up of 5-min low intensity cycling on a cycle ergometer, followed by whole body stretching before any testing or training. Before the pre-test, a single leg squat test was conducted for each leg to select the less stable limb, which was subsequently tested and trained.

After the pre-test session, participants were assigned either to the eccentric or the concentric hip abduction exercise training experimental groups following a matched group design based on the pre-testing data: we applied the z-score method, which is a direct group allocation procedure. That is, we calculated the z-scores and ranked the participants based on the maximal voluntary eccentric torque, DKV values for the two types of jumps, and single leg CMJ propulsive mechanical impulse. As a result, we created two groups for the measured variables (between-group difference in eccentric torque: $p = 0.88$, single leg CMJ DKV: $p = 0.59$, single leg landing DKV: $p = 0.66$, and single leg CMJ propulsive mechanical impulse: $p = 0.57$) for the eccentric and the concentric training groups (Table 1). Thus, each participant in the eccentric group had her matched pair in the concentric group with nearly identical abilities. Within these pairs, we equalized the mechanical work achieved in the eccentric and the concentric hip abductions during the entire training period. The post-test was conducted with the same measurements used in the pre-test, 72 hours after the last training session. Participants

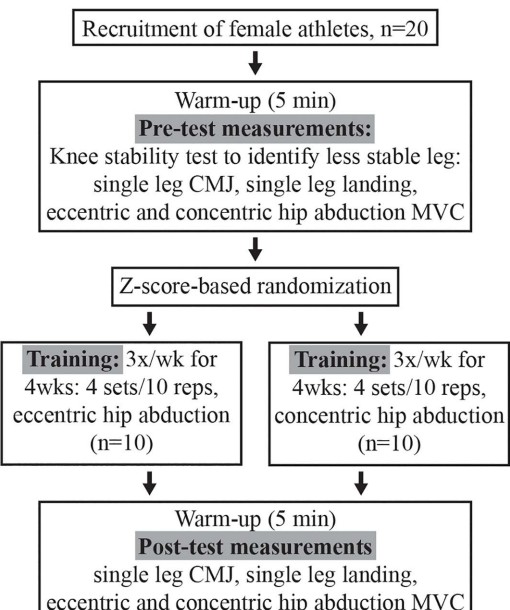

**Fig 1. Flowchart of the study.** CMJ = countermovement jump, MVC = maximal voluntary contraction.

**Table 1. Participants' group-specific descriptive data.**

|  | EC group (n = 10) | CC group (n = 10) | p |
|---|---|---|---|
| Age (years) | 21.9 ± 1.7 | 20.8 ± 3.1 | 0.405 |
| Height (cm) | 167.1 ± 5.5 | 168.5 ± 6.3 | 0.650 |
| Body mass (kg) | 60.4 ± 5.6 | 60.8 ± 7.0 | 0.907 |
| Years in sports (years) | 15.9 ± 3.10 | 14.9 ± 3.6 | 0.948 |
| Training (hour/week) | 6.2 ± 2.4 | 6.3 ± 2.6 | 0.697 |

Data are mean ±SD.

EC = training with eccentric contractions.

CC = training with concentric contractions.

were unaware about the identity of their matched partner in the other experimental group. Also, participants were informed about their individual improvements only after the completion of the study.

## Maximal voluntary eccentric hip abduction contraction test

We measured the maximum voluntary hip abduction eccentric and concentric torque using a computer-controlled isokinetic dynamometer with a sampling frequency of 1.0 kHz (Multicont II, Mediagnost, Budapest and Mechatronic Ltd, Szeged, Hungary). Participants were positioned on their side opposite of the tested leg on the dynamometer. The lever arm starting position was set at 23° frontal-plane hip angle, and a constant velocity of 10°/s was used for eccentric measurements. Participants resisted the lever arm with maximal force until it reached the final position of 0°. The lever arm starting position was set at 0° frontal-plane hip angle, and a constant velocity of 10°/s was used for concentric measurements. Participants resisted the lever arm with maximal force until it reached the final position of 23°. Prior to testing, participants executed two submaximal warm-up trials at ~60 and ~80% of the estimated maximum effort. Then, two maximal effort contractions were executed with two minutes of inter-trial rest. Peak torque values were recorded, and the higher value was used for further analyses. To ensure maximal effort, participants received verbal encouragement.

## Single leg CMJ

Participants performed the single leg CMJ test on a force platform (Tenzi Ltd., Pilisvörösvár, Hungary). Standing on one leg, with arms akimbo and the knee of the free leg slightly flexed, participants performed three jumps with one minute of inter-trial rest. Participants were instructed to jump as high as possible with hands kept on the hips, but no instructions were given on jumping strategy. We recorded the vertical ground reaction force at a sampling rate of 420 Hz. The propulsive mechanical impulse was then calculated from the force-time curve and the impulse values normalized for body mass.

## Single leg drop landing

For the single leg landing test, participants stepped off from a 46 cm-high plyometric box (Capital Sports Plyo Crew Box) and landed on the force plate with the stepping leg. Participants were instructed to step off the box without jumping up and after landing on one leg, hold the landing position for 5 s. Participants performed three trials with 2 min of inter-trial rest.

## Knee joint kinematics

A 3D motion tracking system (Noraxon, Scottdale, USA, sampling frequency: 100 Hz) was used to analyze the kinematics of single leg CMJs and single leg drop landings. Sensors comprising accelerometer-gyroscope-magnetometer complex were affixed with Velcro straps on the shank and the thigh. We followed the manufacturer's recommendations in sensor

positioning and calibration. The sensors provided data to compute DKV from frontal plane orientation angle and maximum knee flexion in the sagittal plane during single leg CMJs and landings.

### Hip abduction exercise training on an isokinetic dynamometer

Participants trained 3 times/week for four weeks on the same dynamometer used for testing (Multicont II, Mediagnost, Budapest and Mechatronic Ltd, Szeged, Hungary). To control for mechanical load, we used a mechanical work-matched design [32–35]. In a session, both the eccentric and the concentric group performed 4 sets of 10 maximal effort repetitions of eccentric and concentric hip abduction contractions with 5 min of inter-set rest. The range of motion and the contraction velocity were identical to those used in the testing protocol described above. Because in a concentric-eccentric subject pair the eccentric subject was able to do more mechanical work, concentric subjects performed an additional set (set 5 with individual repetition adjustments) to equate mechanical work in the two groups. This means, that within a subject pair, the eccentric subject did the given training session (4 sets of 10 reps) first, and her total mechanical work was calculated immediately, using the dynamometer's data output for every contraction. Afterwards, her concentric pair performed the session also with 4 sets of 10 reps, for which, we also calculated the total mechanical work. Using then the total work difference between a concentric and an eccentric subject in a session, we estimated how many more repetitions the concentric subject needed to perform in set 5. This procedure was administered in every training session, and the total load in terms of mechanical work for the four-week training, therefore, was identical in the two groups (see Results).

### Data processing and statistical analyses

We present the data as mean ± standard deviation (SD). We checked each variable with the Shapiro-Wilk test for normality. When a variable revealed not normal distribution, we log-transformed the data for analyses, but we report the non-transformed data in Tables and Figures. The key analysis was a group (eccentric, concentric) by time (pre, post) analysis of variance (ANOVA) with repeated measures on Time. A significant a group by time interaction was followed with a Tukey's post-hoc contrast to determine the means that differed at $p < 0.05$. We characterize these statistical effects with partial eta squared ($\eta p^2$, F values) and pair-wise time-point and between-group differences with Cohen's d effect sizes.

Cutoffs for $\eta p^2$ are ≥ 0.01 (small), ≥ 0.06 (medium), and ≥ 0.14 (large) [36]. Cutoffs for d are small = 0.20; moderate = 0.50; large = 0.80 [36].

Statistical power was calculated post hoc (G*Power 3.1.9.7), and the tests revealed very high statistical power values: DKV during single leg CMJ = 1.0, DKV during single leg landing = 0.99, eccentric hip abduction torque = 1.0, concentric hip abduction torque = 1.0.

The level of significance was set at $p < 0.05$.

## Results

All variables were normally distributed. The two training groups did not differ at baseline in any of the variables (all $p > 0.05$). The total mechanical work performed by participants during the 4 weeks of eccentric (23,319 ± 4,713 J) and concentric (23,117 ± 3,781 J) hip abduction training did not differ (p = 0.458).

Table 2 shows the pre- and post-exercise data. There was a Time main effect for eccentric hip abduction torque (F = 39.7, $\eta p^2$ = 0.700 $p < 0.001$) without a group by time interaction. Similarly, there was a Time main effect for concentric hip abduction torque (F = 26.7, $\eta p^2$ = 0.611, $p < 0.001$) without a group by time interaction.

DKV decreased in single leg CMJ and single leg drop landing (Time main effect: F = 33.5, $\eta p^2$ = 0.663, $p < 0.05$ and F = 14.80, $\eta p^2$ = 0.465, $p < 0.05$). In the single leg CMJ test, there also was a group by time interaction for DKV (F = 6.57, $\eta p^2$ = 0.279, $p < 0.05$). The post-hoc analysis revealed that DKV decreased more in the eccentric vs. concentric group ($p < 0.05$).

**Table 2. Effects of 12 sessions of eccentric contraction and concentric contraction hip abductor strength training on eccentric hip abduction torque, concentric hip abduction torque, dynamic knee valgus, knee flexion, and single-leg CMJ mechanical impulse.**

| Variables | Group | Pre test | Post test | Δ, abs | Δ, % | d | ANOVA | | | |
|---|---|---|---|---|---|---|---|---|---|---|
| | | | | | | | Time main effect | | Group x Time interaction | |
| | | | | | | | F | p | F | p |
| Eccentric hip abduction torque (Nm) | EC (n = 10) | 122.8 ± 16.47 | 147.7 ± 35.59 | 24.89 | 20.27 | 0.84 | 39.7 | 0.000** | 1.2 | 0.284 |
| | CC (n = 10) | 124.4 ± 24.33 | 159.9 ± 25.65 | 35.50 | 28.54 | 1.51 | | | | |
| Concentric hip abduction torque (Nm) | EC (n = 10) | 109.10 ± 19.96 | 144,70 ± 31.23 | 35.60 | 32.63 | 1.37 | 26.7 | 0.000** | 0.48 | 0.5 |
| | CC (n = 10) | 110.30 ± 27.72 | 156.80 ± 40.01 | 46.50 | 42.16 | 1.39 | | | | |
| DKV during SLCMJ (°) | EC (n = 10) | 19.1 ± 10.17 | 10.6 ± 9.20 | −8.56 | −44.79 | 0.94 | 33.5 | 0.000** | 5,57 | 0.016* |
| | CC (n = 10) | 17.2 ± 6.35 | 12.1 ± 6.72 | −5.06 | −29.49 | 0.82 | | | | |
| DKV during SLL(°) | EC (n = 10) | 20.1 ± 14.39 | 4.4 ± 4.67 | −15.70 | −78.11 | 1.27 | 14.8 | 0.001* | 3.9 | 0.066 |
| | CC (n = 10) | 20.9 ± 10.47 | 13.2 ± 9.04 | −7.70 | −36.84 | 0.83 | | | | |
| SLCMJ knee flexion (°) | EC (n = 10) | 81.2 ± 8.77 | 79.3 ± 5.21 | −1.90 | −2.34 | 0.26 | 0.0 | 0.93 | 1.1 | 0.329 |
| | CC (n = 10) | 82.7 ± 4.51 | 84.4 ± 4.98 | 1.75 | 2.12 | 0.39 | | | | |
| I SLCMJ, Ns·kg⁻¹ | EC (n = 10) | 0.3 ± 0.054 | 0.4 ± 0.07 | 0.03 | 9.09 | 0.49 | 5.1 | 0.039* | 0.1 | 0.723 |
| | CC (n = 10) | 0,3 ± 0.05 | 0.4 ± 0.04 | 0.02 | 5.88 | 0.46 | | | | |

Values are mean ±SD.

EC, hip abduction training using eccentric muscle contraction.

CC, hip abduction training using concentric muscle contraction.

DKV, dynamic knee valgus.

SLCMJ, single leg countermovement jump.

SLL, single leg landing.

I, impulse.

Δ abs, absolute change.

Δ %, percent change.

d, Cohen's effect size.

*, p < 0.05.

**, p < 0.005.

Regarding the changes in DKV measured during single leg drop landings, the group by time interaction was not significant (F = 3.9, $\eta p^2$ = 0.185).

The training interventions had no effects on maximum knee flexion measured during single leg CMJ (p > 0.05).

The training interventions induced similar improvements in the mechanical impulse measured during single leg CMJ (F = 5.08, $\eta p^2$ = 0.241, p = 0,039), without group by time interaction.

## Discussion

We examined the effects of eccentric and concentric hip abduction training on DKV measured during single leg jumping and drop landing in young asymptomatic females. We found that high-intensity eccentric hip abductor strength training improved DKV in a task-dependent manner in asymptomatic young women. That is, eccentric vs. concentric training reduced DKV in single leg CMJ more than in single leg drop landing. While both interventions improved single leg CMJ impulse, neither modified knee flexion during single leg CMJ.

### Effects of hip abductor strength training on hip abduction torque

Eccentric and concentric hip abduction training induced favorable but similar adaptations in eccentric hip abduction torque (20% and 28%, or 1.7% and 2.3% per session, respectively), as well as in concentric hip abduction torque (33% and 42%, or 2.7% and 3.5% per session, respectively). These changes fail to support Hypothesis 1, as there was a lack of training-specificity with respect to contraction mode. Improved hip abduction torque following hip abduction resistance training has been observed previously, with improvements varying between 0.8 to 41.0% from pre- to post-intervention, or 0.04 to 2.06% improvements per session. [18,22,37]. While the torque increases we observed are within this range, we note that previous studies used an isometric hip abduction test contraction.

To our knowledge, this is the first study reporting adaptation in eccentric hip abduction torque after pure eccentric and concentric hip abduction strength training. The lack of group by time interaction for eccentric hip abduction torque can perhaps be explained with our dynamometric exercise protocol: the range of motion during training was set to a narrow 23°, which perhaps prevented contraction-specific adaptations in torque generation to emerge. It is also possible that training with either contraction mode was unusual so that performing these unaccustomed contractions produced similar neural adaptations, underlying the torque changes.

### Effects of hip abductor strength training on DKV

Internal rotation of the tibia and high values of DKV seem to play a significant role in the mechanism of ACL tears [38]. One effective way to decrease DKV during single-leg movements is by activating the hip abductor muscles to abduct the femur [3]. Consequently, having enough tension in the hip abductor muscles during single leg countermovement jumps and single leg landings can serve as a preventive measure against large DKV, decreasing the risk of an ACL tear. Hip abduction in resistance exercises can effectively reduce DKV when performing movements with a single leg. However, it should be noted that the DKV in those studies was measured using a single-leg squat test, which may not accurately capture the fast movements that can result in ACL tear [17,18]. In our study, maximal effort eccentric and concentric training each reduced DKV during single leg CMJ so that the reductions were greater after eccentric ($-45\%$, $d = 0.94$) than concentric ($-30\%$, $d = 0.82$). Such differential effects were not present in DKV when measured during single leg landing after training ($-58\%$, $d = 0.88$, Table 2). These data partially support hypothesis 2.

One strategy to minimize the risk of severe knee injuries during single-leg movements is to consciously prepare and regulate the extent of DKV. Previous studies found either small [17] or no effects [18] of hip abduction exercise on DKV, or when favorable effects occurred, a low-velocity squat was used as a test [17,18,20], though making direct comparisons are difficult as we used different population in the current study. To our knowledge, the present study is the first that used eccentric hip abductor strength training to reduce DKV. Because there was a lack of specificity in the training adaptations (discussed previously) while eccentric strength training still induced greater reductions in DKV during single leg CMJ, we interpret these data to mean that participants learned to activate the hip abductors eccentrically. In other words, regardless of the training type (eccentric, concentric), strength gains could be more effectively utilized during an eccentric task. Why such ability was present only in single leg CMJ but not in single leg landing, remains to be further examined. The inconsistency between single leg CMJ and single leg landing adaptations might be related to differences between these

two tasks in various neurokinematical mechanisms. First, we previously found that participants activated their gluteus medius muscle in a single leg CMJ 60% more than in a landing task, though our data is limited to males [39]. Difference in knee flexion strategy is also evident, showing that both males and females use considerably smaller knee flexions during landing compared with CMJ [40,41]. This latter suggests that DKV valgus develops faster in landing, and perhaps eccentric hip abductor training with faster contractions would reveal greater reductions in DKV. The neurokinematical differences between CMJ and landing altogether might be the reason that the two training modalities in the current study did not differentially affect the landing task.

### Effects of hip abductor strength training on single leg CMJ performance

We found that improvements in single leg CMJ mechanical impulse did not differ after eccentric (9%) and concentric (6%) strength training. There is limited information on the effects of hip abductor strength training on vertical jump performance. Our preliminary data strengthens the report that 6 weeks of hip abduction training increased vertical jump performance by 11% in female volleyball players [42]. Giustino et al. reviewed in detail the potential mechanisms of how hip abduction training could improve vertical jump performance. Briefly, a standard inverse dynamics analysis revealed identical hip (31%), knee (34%), and ankle (35%) joint power contribution to leg CMJ propulsion, suggesting that any dysfunction in these joints would negatively affect vertical jump performance [43]. Furthermore, tibial-femoral misalignment can induce atypical loading patterns leading to reduced jump performance in volleyball players [44]. The above mechanisms together with the fact that hip abductor muscles control DKV [13–15] generally explain jump performance improvements after hip abductor strengthening.

In the present study, we found no superior effects of eccentric vs. concentric hip abductor strength training on single leg CMJ mechanical impulse, rejecting Hypothesis 3. Optimal femur alignment can enhance CMJ performance, and despite that our eccentric vs. concentric strength training reduced DKV more (see discussion below), the better femur position still seems insufficient to favor single leg CMJ performance. Given that the role of anterior and posterior leg muscle strength in single leg CMJ performance is beyond doubt [45], it can be still inferred that hip abductor strength training also has a notable impact on single leg CMJ performance.

### Effect of hip abductor strength training on knee flexion during single leg CMJ

Because women tend to land with small knee flexion to avoid excessive DKV, we proposed that hip abductor strengthening would change knee flexion strategy during single leg CMJ. This was, however, not the case because neither eccentric nor concentric hip abduction training did modify knee flexion during single leg CMJs. It is also possible that our participants had weak knee extensor mechanism to generate impulse for single leg CMJs, preventing jumps executed with a greater knee flexion, producing higher single leg CMJ performance. Our participants used only 79–84° knee flexion, even though untrained female college students were reported to used 90° knee flexions during single leg CMJ [46]. It thus seems that changes in knee flexion did not contribute to the improved single leg CMJ performance.

### Limitations, conclusion

Future studies could address the following limitations. We performed the present study in healthy young recreational females with DKV and such data may still not be relevant to high-level athletes. We did not measure muscle activation in the six hip abductor muscles during training and jumping and it is possible that these muscles activated differently during the training tasks and the test tasks. Our biomechanical analyses did not include inverse dynamics or musculoskeletal modeling to provide mechanistic insights into how training modifies DKV and jumping performance. Finally, training the hip abductors in a side-lying, non-weight-bearing position is quite different from the functional, weight-bearing context in which DKV occurs (e.g., jumping and landing). Therefore, the impressive strength gains (20–42%) may be specific to the testing/training apparatus and may not fully translate to the coordinated, multi-joint control required during the single-leg CMJ

and landing tasks. In conclusion, high-intensity eccentric versus concentric hip abductor strength training was superior in dynamic knee valgus improvement measured during single-leg countermovement jump but not during drop landings, while both training modalities improved single leg countermovement jump performance in asymptomatic young women.

## Author contributions

**Conceptualization:** Ádám Fésüs, Balázs Sebesi, Judit Prókai, Kitty Vadász, Balázs Gáspár, Márk Váczi.

**Data curation:** Ádám Fésüs, Tibor Hortobágyi, Balázs Sebesi.

**Formal analysis:** Patrik Ivusza.

**Investigation:** Ádám Fésüs, Balázs Sebesi, Patrik Ivusza, Judit Prókai, Kitty Vadász, Balázs Gáspár, Vanessza Malmos.

**Methodology:** Ádám Fésüs, Zsolt Murlasits, Patrik Ivusza, Judit Prókai, Kitty Vadász, Balázs Gáspár, Vanessza Malmos, Márk Váczi.

**Project administration:** Ádám Fésüs, Vanessza Malmos, Márk Váczi.

**Supervision:** Tibor Hortobágyi, Márk Váczi.

**Validation:** Márk Váczi.

**Visualization:** Balázs Sebesi.

**Writing – original draft:** Ádám Fésüs, Tibor Hortobágyi, Balázs Sebesi, Zsolt Murlasits, Patrik Ivusza, Márk Váczi.

**Writing – review & editing:** Tibor Hortobágyi, Zsolt Murlasits, Márk Váczi.

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
