## [Decision Letter · Decision Letter 0]

1 Dec 2025

PONE-D-25-39091High-intensity eccentric hip abductor strength training improves dynamic knee valgus in a task-dependent manner in asymptomatic young women: a pilot studyPLOS ONE

Dear Dr. Fésüs,

Thank you for submitting your manuscript to PLOS ONE. After careful consideration, we feel that it has merit but does not fully meet PLOS ONE’s publication criteria as it currently stands. Therefore, we invite you to submit a revised version of the manuscript that addresses the points raised during the review process.

If applicable, we recommend that you deposit your laboratory protocols in protocols.io to enhance the reproducibility of your results. Protocols.io assigns your protocol its own identifier (DOI) so that it can be cited independently in the future. For instructions see: https://journals.plos.org/plosone/s/submission-guidelines#loc-laboratory-protocols. Additionally, PLOS ONE offers an option for publishing peer-reviewed Lab Protocol articles, which describe protocols hosted on protocols.io. Read more information on sharing protocols at . Additionally, PLOS ONE offers an option for publishing peer-reviewed Lab Protocol articles, which describe protocols hosted on protocols.io. Read more information on sharing protocols at https://plos.org/protocols?utm_medium=editorial-email&utm_source=authorletters&utm_campaign=protocols..

We look forward to receiving your revised manuscript.

Kind regards,

Nili Steinberg

Academic Editor

PLOS ONE

[ÁF, PI, and BG was supported by the Ildikó Kriszbacher scholarship, University of Pécs. BS was supported from EKÖP-24-4-I-PTE-379, funded by the Ministry of Culture and Innovation, National Fund for Research, Development and Innovation, under the University Research Grant Programme EKÖP-24-1. For the remaining authors none were declared.].

Additional Editor Comments (if provided):

Reviewers' comments:

Reviewer's Responses to Questions

**Comments to the Author**

1. Is the manuscript technically sound, and do the data support the conclusions?

Reviewer #1: Yes

Reviewer #2: Partly

2. Has the statistical analysis been performed appropriately and rigorously? 

Reviewer #1: Yes

Reviewer #2: I Don't Know

3. Have the authors made all data underlying the findings in their manuscript fully available?

Reviewer #1: Yes

Reviewer #2: Yes

4. Is the manuscript presented in an intelligible fashion and written in standard English?

Reviewer #1: Yes

Reviewer #2: Yes

5. Review Comments to the Author

Reviewer #1: This is an automated report for PONE-D-25-39091. This report was solicited by the PLOS One editorial team and provided by ScreenIT.

ScreenIT is an independent group of scientists developing automated tools that analyze academic papers. A set of automated tools screened your submitted manuscript and provided the report below. Each tool was created by your academic colleagues with the goal of helping authors. The tools look for factors that are important for transparency, rigor and reproducibility, and we hope that the report might help you to improve reporting in your manuscript. Within the report you will find links to more information about the items that the tools check. These links include helpful papers, websites, or videos that explain why the item is important. While our screening tools aim to improve and maintain quality standards they may, on occasion, miss nuances specific to your study type or flag something incorrectly. Each tool has limitations that are described on the ScreenIT website. The tools screen the main file for the paper; they are not able to screen supplements stored in separate files. Please note that the Academic Editor had access to these comments while making a decision on your manuscript. The Academic Editor may ask that issues flagged in this report be addressed. If you would like to provide feedback on the ScreenIT tool, please email the team at ScreenIt@bih-charite.de. If you have questions or concerns about the review process, please contact the PLOS One office at plosone@plos.org.

Reviewer #2: Hello to all authors,Was this pilot study conducted to assess participants’ acceptance of the protocol, to evaluate the validity of measurement tools, or to estimate the dropout rate for a future main study?

6. PLOS authors have the option to publish the peer review history of their article (what does this mean?). If published, this will include your full peer review and any attached files.). If published, this will include your full peer review and any attached files.

.

Reviewer #1: No

Reviewer #2: **Yes:** Dr. Mohadeseh AshrafizadehDr. Mohadeseh Ashrafizadeh

---

## [Author Response · Author response to Decision Letter 1]

6 Jan 2026

RESPONSES TO EDITOR AND REVIEWERS

Manuscript ID: PONE-D-25-39091

Dear Editor,

Thank you for handling the reviewing process and for the comments. Please find below our point-by-point responses:

Response to the Editor’s comments:

Comment 1

Response to comment 1

The manuscript has been formatted according to the linked formatting criteria.

Comment 2

Response to comment 2

Funders had no role in study design, data collection and analysis, decision to publish, or preparation of the manuscript. We also make this statement in the cover letter.

Comment 3

We note that your Data Availability Statement is currently as follows: [All relevant data are within the manuscript and its Supporting Information files.]

Response to comment 3

The minimal dataset required is available on the Zenodo platform under the following DOI:https://doi.org/10.5281/zenodo.1808373 .

This access location is indicated in the Data Availability Statement section of the manuscript, between lines 395–398.

Comment 4

PLOS requires an ORCID iD for the corresponding author in Editorial Manager on papers submitted after December 6th, 2016. Please ensure that you have an ORCID iD and that it is validated in Editorial Manager. To do this, go to ‘Update my Information’ (in the upper left-hand corner of the main menu), and click on the Fetch/Validate link next to the ORCID field. This will take you to the ORCID site and allow you to create a new iD or authenticate a pre-existing iD in Editorial Manager.

Response to comment 4

Thank you for calling attention. We have created the ORCID ID for the corresponding author.

ORCID iD: 0009-0007-2246-0883

ORCID record: https://orcid.org/0009-0007-2246-0883

Comment 5

Response to comment 5:

The reviewers had no recommendations to cite specific works.

Response to reviewers’ comments

Responses to ScreenIT report (Reviewer 1) comment

Comment 1:

Flow charts and attrition

We did not find a study flow chart of excluded observations. We strongly recommend using flow charts because they provide an overview of the study design and more information about attrition. If you included a study flow chart in your supplemental files, we apologize for missing this. Our tool is not able to screen separate supplemental files.

Sentence about attrition: not detected. Please provide information about the drop-out of subjects, or loss of animals or samples. This could be done using a study flow chart, or described in the text.

Response to comment 1

The study flow chart was already uploaded among the submitted files as Figure 1, presenting the participant inclusion and exclusion process.

There was no attrition during the study, and we state this in the Materials and Methods section on line 130.

Comment 2

Randomization

Not detected. If you performed an experimental study, please specify whether participants, animals or samples were randomly assigned to treatments or groups, and specify the randomization procedure.

Response to comment 2

We applied the Z-score method, which is a directed group allocation procedure. The description of this method can be found in lines 142-144 of the manuscript. We made small modifications in the manuscript text. The essence of our method is to ensure that the baseline values of the measured variables did not differ between the two training groups. Statistical results showing the comparison of the baseline values are presented in lines 144-147).

Comment 3

Blinding

Not detected. Please specify whether blinding was used at various phases of the experiment (e.g., blinding of patients, caregivers, outcome assessments, data analysis).

Response to comment 3

We have supplemented the Experimental Procedure session with the following sentences: „Participants were unaware about the identity of their matched partner in the other experimental group. Also, participants were informed about their individual improvements only after the completion of the study

Comment 4

Power Analysis

Not detected. Please state how you determined the sample size for the study. If you performed a power or sample size calculation, please provide details.

Response to comment 4

Because this was a pilot study, according to the journal’s policy, formal power calculation was not required.

Comment 5

Open code

If you wrote code to analyze your data, please consider sharing the code in a public repository. This makes it easier for others to reproduce your analyses, and may also aid others seeking to analyze similar datasets.

Response to comment 5

We did not develop any specific computer program and did not use any computer codes to reduce and analyse data. We used cursor-driven methods to manually analyse the data. For statistical followed only standards statistical procedures.

Comment 6

Open data

If permitted, sharing data on a public repository with a persistent identifier (a DOI or accession number) can improve reproducibility and make it easier for other scientists to expand on your work. Papers with open data are cited more often than papers without open data. Some institutions have an expert who can provide advice on data sharing.

Response to comment 6

As noted above under ‘Response to Editor’s comment 3’, we have uploaded an excel sheet with the raw data in the Supporting Information Files

Responses to reviewer 2

Comment 1

Hello to all authors. Was this pilot study conducted to assess participants’ acceptance of the protocol, to evaluate the validity of measurement tools, or to estimate the dropout rate for a future main study?

Response to comment 1

The study aimed to compare the effects of a four-week-long dynamometric concentric vs. eccentric hip abduction exercise training on eccentric hip abduction peak torque, magnitude of dynamic knee valgus, and single leg counter-movement jump mechanical impulse in healthy young females diagnosed with dynamic knee valgus. Therefore, this study was not designed specifically to assess participants’ acceptance of the protocol, to evaluate the validity of measurement tools, or to estimate the dropout rate for a future main study. We note that after enrolment, no participant has dropped out, implying 100% acceptance of the protocols and 0% dropout. Results from this pilot study offer valuable preliminary evidence supporting the extension of the protocol into the clinical field.

---

## [Decision Letter · Decision Letter 1]

18 Mar 2026

PONE-D-25-39091R1High-intensity eccentric hip abductor strength training improves dynamic knee valgus in a task-dependent manner in asymptomatic young women: a pilot studyPLOS One

Dear Dr. Fésüs,

Thank you for submitting your manuscript to PLOS ONE. After careful consideration, we feel that it has merit but does not fully meet PLOS ONE’s publication criteria as it currently stands. Therefore, we invite you to submit a revised version of the manuscript that addresses the points raised during the review process.

If applicable, we recommend that you deposit your laboratory protocols in protocols.io to enhance the reproducibility of your results. Protocols.io assigns your protocol its own identifier (DOI) so that it can be cited independently in the future. For instructions see: https://journals.plos.org/plosone/s/submission-guidelines#loc-laboratory-protocols. Additionally, PLOS ONE offers an option for publishing peer-reviewed Lab Protocol articles, which describe protocols hosted on protocols.io. Read more information on sharing protocols at . Additionally, PLOS ONE offers an option for publishing peer-reviewed Lab Protocol articles, which describe protocols hosted on protocols.io. Read more information on sharing protocols at https://plos.org/protocols?utm_medium=editorial-email&utm_source=authorletters&utm_campaign=protocols..

We look forward to receiving your revised manuscript.

Kind regards,

Mário Espada, PhD

Academic Editor

PLOS One

**Journal Requirements:**

**Additional Editor Comments:**

Dear Authors,

Congratulations on your work.

Please revise the manuscript in light of the reviewers' minor suggestions.

Thank you.

Best regards.

Reviewers' comments:

Reviewer's Responses to Questions

**Comments to the Author**

1. If the authors have adequately addressed your comments raised in a previous round of review and you feel that this manuscript is now acceptable for publication, you may indicate that here to bypass the “Comments to the Author” section, enter your conflict of interest statement in the “Confidential to Editor” section, and submit your "Accept" recommendation.

Reviewer #2: (No Response)

Reviewer #3: (No Response)

Reviewer #4: (No Response)

2. Is the manuscript technically sound, and do the data support the conclusions?

Reviewer #2: Yes

Reviewer #3: Yes

Reviewer #4: Partly

3. Has the statistical analysis been performed appropriately and rigorously? 

Reviewer #2: I Don't Know

Reviewer #3: Yes

Reviewer #4: (No Response)

4. Have the authors made all data underlying the findings in their manuscript fully available?

Reviewer #2: Yes

Reviewer #3: Yes

Reviewer #4: Yes

5. Is the manuscript presented in an intelligible fashion and written in standard English?

Reviewer #2: Yes

Reviewer #3: Yes

Reviewer #4: Yes

6. Review Comments to the Author

Reviewer #2: (No Response)

Reviewer #3: Reviewer Comments on Manuscript PONE-D-25-39091

Title: High-intensity eccentric hip abductor strength training improves dynamic knee valgus in a task-dependent manner in asymptomatic young women: a pilot study

Journal: PLOS ONE

Reviewer’s Assessment: Major Revisions Recommended

General Overview:

This pilot study investigates a relevant and timely research question: whether the mode of hip abductor strength training (eccentric vs. concentric) differentially affects dynamic knee valgus (DKV) and jump performance in asymptomatic young women. The study is well-motivated, highlighting the often-overlooked eccentric control function of the hip abductors. The use of an isokinetic dynamometer for precise, maximal-effort training and testing is a significant strength. The findings that eccentric training was superior in reducing DKV during a single-leg countermovement jump (SLCMJ) but not during a drop landing are novel and interesting. However, several methodological ambiguities, statistical reporting issues, and overinterpretations in the discussion need to be addressed before the manuscript is suitable for publication.

Major Concerns:

Participant Selection and DKV Quantification:

The inclusion criterion of "presence of DKV subjectively evaluated by a physiotherapist using the single leg squat test" is a major methodological weakness. Subjective visual assessment is known to have limited reliability. To strengthen the study's validity, the authors should provide data on the inter- or intra-rater reliability of this assessment. Furthermore, since baseline DKV values were subsequently measured with 3D motion capture, why were these objective data not used as the primary inclusion criterion? A minimum threshold of DKV (e.g., >10°) would have created a more homogeneous and relevant sample.

Ecological Validity of the Training Intervention:

While the use of an isokinetic dynamometer allows for excellent experimental control, its ecological validity is a concern. Training the hip abductors in a side-lying, non-weight-bearing position is quite different from the functional, weight-bearing context in which DKV occurs (e.g., jumping and landing). The authors should discuss this as a key limitation. The impressive strength gains (20-42%) may be specific to this testing/training apparatus and may not fully translate to the coordinated, multi-joint control required during the SLCMJ and landing tasks.

Statistical Analysis and Interpretation:

Interaction Effects: The key finding of a significant group-by-time interaction for DKV during the SLCMJ (p < 0.05) is well-presented. However, the interpretation of the non-significant interaction for DKV during the landing task (p > 0.05) needs to be more nuanced. The paper states "eccentric vs. concentric training reduced DKV in single leg CMJ more than in single leg drop landing." This is accurate, but it should be emphasized that the training did not differentially affect the landing task. The discussion should explore potential reasons for this task-dependent adaptation more deeply.

Correlation Analysis: The lack of correlation between strength gains and DKV reduction is a critical finding that challenges the study's primary hypothesis. The current discussion offers plausible reasons (e.g., timing of activation vs. maximal strength, differential muscle usage). This section should be expanded, as it points toward neuromuscular control rather than raw strength being the primary driver of the observed improvements. The authors should consider whether the sample size (n=20) was sufficient to detect a moderate correlation, and note this as a limitation for the correlation analysis.

Clarity of Methods - Work Equilibration:

The description of equating mechanical work between groups needs clarification. The authors state that concentric subjects performed an additional 5th set to match the work of the eccentric subjects. This implies that the training stimulus was not identical in volume. It also suggests that the "maximal effort" in the concentric group's 5th set may have been performed under significant fatigue. Please provide more detail on the timing of this 5th set and how the total work was calculated and matched session-by-session.

Minor Concerns:

Terminology and Grammar:

"Dynamic knee valgus": This term is correctly used, but the manuscript would benefit from a brief operational definition early in the methods (e.g., "defined as the frontal plane projection angle...").

"countermovement jump": Please ensure this term is consistently capitalized (CMJ) after its first use, as it appears in lowercase in several places (e.g., Abstract line 40, Results line 256).

Abstract Line 37: "dynamic knee valgus" should be capitalized as "Dynamic knee valgus" to start the sentence.

Data Presentation:

Table 2: The table is very dense. Consider splitting the ANOVA results into a separate table or using symbols (e.g., *, †) directly in the table to denote significant main effects and interactions for easier readability. The column headers for the ANOVA results are also difficult to follow.

Specific Discussion Points:

Lines 307-310: The statement regarding the lack of effects in previous studies should be tempered. The current study uses a different population, training method (isokinetic), and outcome measure, making direct comparisons difficult.

Conclusion: The conclusion in the abstract and main text ("improves dynamic knee valgus and vertical jump performance") is too broad. It should specify that it improves DKV in a task-dependent manner (favoring SLCMJ) and improves jump performance similarly to concentric training.

Questions for the Authors:

What was the rationale for choosing a 4-week intervention? Was a power analysis conducted to determine the sample size of 20, and if so, what was the expected effect size for the primary outcome (DKV change)?

Were the participants blinded to their group allocation (eccentric vs. concentric)? If not, how might this have influenced the results?

Could the "task-dependent" improvement in the SLCMJ be partially explained by a learning effect specific to the jump task, given that the training did not involve a similar plyometric movement?

Reviewer #4: I suggest softening the conclusions and avoiding statements that imply injury-prevention benefits. Considering the pilot design, the small sample size, the use of DKV as a surrogate outcome, and the lack of muscle activation measures, the results should be interpreted as suggestive findings showing a task-specific effect on DKV rather than evidence of reduced ACL injury risk.

7. PLOS authors have the option to publish the peer review history of their article (what does this mean?). If published, this will include your full peer review and any attached files.). If published, this will include your full peer review and any attached files.

.

Reviewer #2: No

Reviewer #3: No

Reviewer #4: **Yes:** Seham Abdallah ElazabSeham Abdallah Elazab

---

## [Author Response · Author response to Decision Letter 2]

8 Apr 2026

We thank the Editor and the Reviewers for their constructive comments and positive feedback. We have provided a detailed, point-by-point response to all reviewer and editor comments in the uploaded “Response to Reviewers” document. All comments have been addressed and the manuscript has been revised accordingly. The revised manuscript now includes all relevant statistical data, including previously reported effect sizes and the newly added power values requested by the reviewers. Together with the raw data repository, this ensures that the study is fully transparent and citable for future investigations and meta-analyses.

---

## [Decision Letter · Decision Letter 2]

16 Apr 2026

High-intensity eccentric hip abductor strength training improves dynamic knee valgus in a task-dependent manner in asymptomatic young women: a pilot study

PONE-D-25-39091R2

Dear Dr. Ádám Fésüs,

We’re pleased to inform you that your manuscript has been judged scientifically suitable for publication and will be formally accepted for publication once it meets all outstanding technical requirements.

An invoice will be generated when your article is formally accepted. Please note, if your institution has a publishing partnership with PLOS and your article meets the relevant criteria, all or part of your publication costs will be covered. Please make sure your user information is up-to-date by logging into Editorial Manager at Editorial Manager® and clicking the ‘Update My Information' link at the top of the page. For questions related to billing, please contact  and clicking the ‘Update My Information' link at the top of the page. For questions related to billing, please contact billing support..

Kind regards,

Mário Espada, PhD

Academic Editor

PLOS One

---

## [Editor Report · Acceptance letter]

PONE-D-25-39091R2

PLOS One

Dear Dr. Fésüs,

I'm pleased to inform you that your manuscript has been deemed suitable for publication in PLOS One. Congratulations! Your manuscript is now being handed over to our production team.

Kind regards,

on behalf of

Dr. Mário Espada

Academic Editor

PLOS One